# Multi-Omics Identifies Circulating miRNA and Protein Biomarkers for Facioscapulohumeral Dystrophy

**DOI:** 10.3390/jpm10040236

**Published:** 2020-11-19

**Authors:** Christopher R. Heier, Aiping Zhang, Nhu Y Nguyen, Christopher B. Tully, Aswini Panigrahi, Heather Gordish-Dressman, Sachchida Nand Pandey, Michela Guglieri, Monique M. Ryan, Paula R. Clemens, Mathula Thangarajh, Richard Webster, Edward C. Smith, Anne M. Connolly, Craig M. McDonald, Peter Karachunski, Mar Tulinius, Amy Harper, Jean K. Mah, Alyson A. Fiorillo, Yi-Wen Chen

**Affiliations:** 1Department of Genomics and Precision Medicine, George Washington University School of Medicine and Health Sciences, Washington, DC 20037, USA; HGordish@childrensnational.org (H.G.-D.); afiorillo@childrensnational.org (A.A.F.); 2Center for Genetic Medicine Research, Children’s National Hospital, Washington, DC 20010, USA; AZhang@childrensnational.org (A.Z.); nnguyen@childrensnational.org (N.Y.N.); ctully2@childrensnational.org (C.B.T.); APANIGRAHI@childrensnational.org (A.P.); spandey@childrensnational.org (S.N.P.); 3Newcastle Upon Tyne Hospitals, Newcastle NE1 3BZ, UK; michela.guglieri@newcastle.ac.uk; 4The Royal Children’s Hospital, Melbourne University, Parkville, Victoria 3052, Australia; monique.ryan@rch.org.au; 5Department of Neurology, University of Pittsburgh School of Medicine, Pittsburgh, PA 15261, USA; pclemens@pitt.edu; 6Department of Neurology, Virginia Commonwealth University School of Medicine, Richmond, VA 23298, USA; mathula.thangarajh@vcuhealth.org; 7Children’s Hospital at Westmead, Sydney 2145, Australia; richard.webster@health.nsw.gov.au; 8Department of Pediatrics, Duke University Medical Center, Durham, NC 27705, USA; edward.smith@duke.edu; 9Nationwide Children’s Hospital, The Ohio State University, Columbus, OH 43205, USA; anne.connolly@nationwidechildrens.org; 10Department of Physical Medicine and Rehabilitation, University of California at Davis Medical Center, Sacramento, CA 95817, USA; cmmcdonald@ucdavis.edu; 11Department of Neurology, University of Minnesota, Minneapolis, MN 55455, USA; karac001@umn.edu; 12Department of Pediatrics, Gothenburg University, Queen Silvia Children’s Hospital, 41685 Göteborg, Sweden; mar.tulinius@vgregion.se; 13Department of Neurology, Virginia Commonwealth University, Richmond, VA 23298, USA; amy.harper@vcuhealth.org; 14Deparment of Pediatrics and Clinical Neurosciences, Cumming School of Medicine, University of Calgary, T2N T3B, Calgary, AB 6A81N4, Canada; Jean.Mah@albertahealthservices.ca

**Keywords:** FSHD, biomarkers, miRNA, proteomics, calprotectin, dystrophy, muscle

## Abstract

The development of therapeutics for muscle diseases such as facioscapulohumeral dystrophy (FSHD) is impeded by a lack of objective, minimally invasive biomarkers. Here we identify circulating miRNAs and proteins that are dysregulated in early-onset FSHD patients to develop blood-based molecular biomarkers. Plasma samples from clinically characterized individuals with early-onset FSHD provide a discovery group and are compared to healthy control volunteers. Low-density quantitative polymerase chain reaction (PCR)-based arrays identify 19 candidate miRNAs, while mass spectrometry proteomic analysis identifies 13 candidate proteins. Bioinformatic analysis of chromatin immunoprecipitation (ChIP)-seq data shows that the FSHD-dysregulated DUX4 transcription factor binds to regulatory regions of several candidate miRNAs. This panel of miRNAs also shows ChIP signatures consistent with regulation by additional transcription factors which are up-regulated in FSHD (FOS, EGR1, MYC, and YY1). Validation studies in a separate group of patients with FSHD show consistent up-regulation of miR-100, miR-103, miR-146b, miR-29b, miR-34a, miR-454, miR-505, and miR-576. An increase in the expression of S100A8 protein, an inflammatory regulatory factor and subunit of calprotectin, is validated by Enzyme-Linked Immunosorbent Assay (ELISA). Bioinformatic analyses of proteomics and miRNA data further support a model of calprotectin and toll-like receptor 4 (TLR4) pathway dysregulation in FSHD. Moving forward, this panel of miRNAs, along with S100A8 and calprotectin, merit further investigation as monitoring and pharmacodynamic biomarkers for FSHD.

## 1. Introduction

Facioscapulohumeral muscular dystrophy (FSHD) is an autosomal dominant muscle disorder with no current therapy, a variable prognosis, and complex genetic and molecular mechanisms. FSHD is caused by aberrant expression of *double homeobox 4* (*DUX4*) due to epigenetic changes of the *D4Z4* repeat region at chromosome 4q35 [1,2,3]. Roughly 95% of patients have Type 1 FSHD (FSHD1) due to contraction of the D4Z4 array; a small portion (~5%) of patients have Type 2 FSHD (FSHD2) caused by mutations in *the structural maintenance of chromosomes flexible hinge domain containing 1* (*SMCHD1*) gene, the *DNA methyltransferase 3B (DNMT3B)* gene, or the *ligand-dependent nuclear receptor-interacting factor 1 (LRIF1)* gene [4,5,6]. The aberrant expression of DUX4 protein causes mis-regulation of genes involved in germline function, oxidative stress responses, myogenesis, post-transcriptional regulation, and additional cellular functions [7,8,9,10,11,12,13]. These downstream molecular changes are believed to cause FSHD, although the exact mechanisms are not clear.

Although the onset of FSHD is generally around adolescent years, a small portion (~4%) of patients present with an early-onset or infantile form of FSHD [14]. Previous studies have shown that the disease severity of FSHD1 is negatively correlated with the size of *D4Z4* repeats [15,16]. Individuals with early-onset FSHD1 tend to have smaller *D4Z4* repeats and more severe disease phenotypes, including more profound muscle weakness, younger age at loss of independent ambulation, and extramuscular manifestations such as retinal vasculopathy or hearing loss [14,15,17,18].

In clinical practice, particularly with pediatric-onset FSHD, there is a low use of serial histological assessments because they require painful biopsies of muscle tissue that typically reveal patchy or uneven pathology. Given this, many patients no longer undergo muscle biopsy once a genetic diagnosis is made. Functional motor scales provide a non-invasive alternative to study neuromuscular disease progression; however, they can show great variability, can be age- or disease stage-limited, and they can be subject to placebo or coaching effects in clinical trials [19,20]. Circulating molecular biomarkers provide a promising alternative to these clinical assessments because they are objective measurements that can be assayed repeatedly over time using minimally invasive methods. Blood-based miRNAs or proteins that measure the progression of disease or a patient response to therapy over time are known as a monitoring biomarker [21]. In clinical trials, monitoring biomarkers may also be used as pharmacodynamic biomarkers to identify patients who are early responders to therapy, to demonstrate exposure-response relationships, or to improve statistical power and modeling. As patient populations are sensitive and limited for this relatively rare pediatric disease, less invasive monitoring or pharmacodynamic biomarkers are important for early-onset FSHD, as frequent serial biopsies are especially problematic in this population.

Recently, circulating miRNAs have emerged as exciting potential diagnostic, prognostic, and drug-responsive biomarkers. This is a class of small non-coding ribonucleic acid (RNA) molecules (~22 nucleotides in length) that can help to regulate gene expression [22], and which are highly stable in biofluids such as blood and urine [23,24]. In rare diseases with highly variable symptoms, such as multiple acyl-coenzyme A dehydrogenase deficiency (MADD), the serum-based detection of muscle-specific miRNAs termed myomiRs can signal the presence of underlying muscle-specific pathologies [25]. In Duchenne and Becker muscular dystrophies, myomiRs are up-regulated in serum from both patient populations, while detection of miR-206 up-regulation can be used to differentially diagnose severe Duchenne versus Becker patients [26,27,28]. In addition to myomiRs, inflammatory miRNAs such as miR-146a, miR-146b, miR-221 and miR-155 have been found to be dysregulated in multiple forms of muscular dystrophies [29,30,31]. These two classes of miRNA show potential as pharmacodynamic biomarkers, with myomiRs proposed for muscle-stabilizing treatments such as gene therapy [32,33], and inflammatory microRNAs proposed for current steroids [34,35] as well as newly emerging dissociative anti-inflammatory drugs such as vamorolone [36,37,38] or edasalonexent [39,40]. In parallel to development of miRNA monitoring biomarkers, new advances in whole exome sequencing are enabling clinicians to diagnose novel mutations in over 60 genes known to be responsible for muscular dystrophies such as FSHD and limb-girdle muscular dystrophy (LGMD) [41,42,43,44]. Together, these advances will help to improve the diagnosis, monitoring, and treatment of a diverse number of diseases affecting muscle.

The development of circulating biomarkers for FSHD has the potential to improve clinical management and to facilitate the development of new treatments. In this study, we test plasma samples from a cohort of individuals with early-onset FSHD1 using both miRNA and proteomic profiling approaches. Our goal is to identify molecules that can be used to monitor FSHD disease activity and that may ultimately facilitate future therapeutic trials. Initial analysis of a discovery group identifies a panel of miRNAs and proteins as biomarker candidates. Bioinformatic analyses of ChIP-seq data provide a rationale for the changes in candidate biomarkers, as their behavior is consistent with changes in transcription factor pathways that are disrupted in FSHD1. Subsequent characterization in separate, non-overlapping groups of FSHD1 patients provides validation of nine biomarkers whose expression can be conveniently assayed by qRT-PCR or Enzyme-Linked Immunosorbent Assay (ELISA), and are increased in early-onset FSHD.

## 2. Materials and Methods

### 2.1. Ethics Statement

We obtained institutional ethics and research review boards approval for these clinical studies from the Institutional Review Board of Children’s National Hospital and at all participating Cooperative International Neuromuscular Research Group (CINRG) sites, in accordance with all requirements, as previously described in Mah et al. 2018 [45]. Written informed consent was obtained from all the participants before the study procedures. Where applicable, informed consent and/or assent was obtained from all patients or legal guardians before enrollment.

### 2.2. Patients and Sample Collection

Plasma samples were collected and biobanked from a previous early-onset FSHD study conducted by CINRG as described by Mah et al. [45]. For the discovery experiments, FSHD1 patients aged 10 to 51 years old were included (*n* = 16 for miRNA discovery, *n* = 25 for proteomics discovery), along with healthy control volunteers (*n* = 8 for miRNA discovery, *n* = 17 for proteomics discovery) aged 16 to 54 years old. All patients had Type 1 FSHD caused by epigenetic changes due to *D4Z4* contraction which results in up-regulation of *DUX4*.

### 2.3. miRNA Profiling

RNA was isolated and quantified from the discovery cohort of patients as described previously [34]. Briefly, RNA was isolated from 150 µL of plasma using Trizol liquid sample (LS) reagent (ThermoFisher, Waltham, MA, USA), then converted to cDNA using the High Capacity Reverse Transcription Kit with multiplexed reverse transcription (RT) primers (ThermoFisher). Synthesized cDNA was then pre-amplified using PreAmp MasterMix with multiplexed TaqMan (TM) primers corresponding to the RT primers used in initial cDNA reaction. Quantitative analysis of miRNA was performed via TaqMan Low-Density Array Cards (TaqMan™ Array Human MicroRNA A Cards v2.0; ThermoFisher). The ThermoFisher Cloud software suite with the Relative quantification (Rq) application was used to perform statistical analysis and determine expression of miRNA in either mild or severe FSHD1 patient groups versus healthy controls. A value > 1 indicates an increase and a value < 1 indicates a decrease in miRNA expression in FSHD1 versus healthy controls, with *p*-values ≤ 0.05 considered significant. To reduce false-positive discovery in this setting, we used an evidence-based approach where candidate miRNAs that significantly increased in the discovery groups were cross-referenced to a separate set of non-overlapping CINRG patients used as a validation group.

### 2.4. Bioinformatics of miRNA Regulation via DUX4 and FSHD-Associated Factors

Surrounding DNA regulatory regions of candidate miRNA genes were queried in ChIP-seq datasets for binding by transcription factors known to be impacted by FSHD. These analyses were performed using the University of California Santa Cruz (UCSC) Genome Browser with alignment to the GECh37/hg19 genome build. For primary effects, due to the underlying mutation that causes FSHD, DUX4 binding was queried. For this, we uploaded a user-supplied DUX4 ChIP-seq track published by Geng et al. [9] to determine which candidate miRNAs displayed physical binding of DUX4 at potential regulatory regions within 100 kb of the gene for each miRNA.

To investigate secondary factors whose dysregulation is associated with FSHD-causing mutations, we investigated DNA binding by transcription factors shown to be significantly up-regulated in cultured human muscle cells using microarray data by Geng et al. [9]. For this, we used ChIP-seq data from the Encyclopedia of DNA Elements (ENCODE) [46,47]. From a master list of DUX4-regulated genes published in [9], we identified a list of 34 transcription factors with ChIP-seq data from ENCODE available within the UCSC Txn Factor ChIP Track and 47 transcription factors from the Txn Factor ChIP E3 Track [48,49,50]. After an initial survey of these full transcription factor lists for the 19 candidate miRNAs, we narrowed down to a shorter focus list of 9 transcription factors whose binding was most frequently associated with the candidate miRNAs. DNA binding by transcription factors was queried in datasets produced using ChIP-seq from all 9 available cell line tracks, including GM12878 (lymphoblasts), H1-hESC (embryonic stem cells), HeLa-S3 (cervical cancer cells), HepG2 (liver cancer cells), HSMM (skeletal muscle myoblasts), HUVEC (umbilical vein endothelial cells), K562 (immortalized myelogenous leukemia cells), NHEK (epidermal keratinocytes), and normal human lung fibroblasts (NHLF).

In addition to binding by DUX4 and the transcription factors described above, ChIP-seq data for histone modifications were queried to gain insight into potential promoter or enhancer regulatory functions for the identified transcription factor binding sites. For this, histone H3K4 tri-methylation (found near promoters), H3K4 mono-methylation (found near regulatory elements), and H3K27 acetylation (found near active regulatory elements) were included. These histone modifications were queried in ChIP-seq datasets using all 9 available cell line tracks.

Pathway analysis was performed using Ingenuity Pathway Analysis software version 52912811. Candidate miRNAs from these studies were uploaded along with transcription factors whose dysregulation is associated with FSHD. Defined network connections were identified using the Pathway Builder application. Molecules confirmed to have established relationships were used to visualize a novel network built from these FSHD expression data.

### 2.5. Expression of Individual miRNAs in a Validation Sample Set

Circulating miRNAs that were significantly up-regulated in individuals affected by FSHD1 were examined in a separate set of non-overlapping CINRG patients used as a validation group. For this group, FSHD patients had a confirmed diagnosis of FSHD1 (*n* = 12; 9 females, 3 males) and were compared to healthy volunteer control samples (*n* = 7; 4 females, 3 males). RNA was isolated from 150 µL of plasma using Trizol LS liquid extraction. Total RNA was converted to cDNA using a High Capacity Reverse Transcription Kit with multiplexed RT primers, pre-amplified using PreAmp MasterMix with multiplexed TM primers, and quantified with individual TaqMan assays on an ABI QuantStudio 7 real time PCR machine (Applied Biosystems; Foster City, CA, USA). Assay IDs used are: miR-32-002109, miR-103-000439, miR-505-002089, miR-146b-001097, miR-29b-000413, miR-34a-000426, miR-141-000463, miR-98-000577, miR-576-3p-002351, miR-9-000583, and miR-142-3p-000464. Expression levels of all miRNAs were normalized to the geometric mean of multiple control genes (miR-150 and miR-342-3p) determined previously to be stable circulating miRNA controls [35,51]. Expression was analyzed in FSHD1 versus healthy control patients via *t*-test analysis, including assessment of directionality. A *p*-value of ≤ 0.05 was considered significant. Data are presented as mean ± SEM unless otherwise noted.

### 2.6. Proteomics Profiling

Plasma samples were first processed using Pierce™ Top 12 Abundant Protein Depletion Spin Columns (Thermo Scientific) before mass spectrometry analyses using the Q Exactive HF mass spectrometer. Briefly, the 12 most abundant proteins from 5 µL of plasma sample were affinity depleted by incubating with Top 12 protein depletion resin. Following this, the unbound fraction was collected according to the manufacturer’s protocol. Proteins were precipitated with pre-cooled acetone (1:5 vol) for 30 min at −20 °C and centrifuged at 4 °C for 15 min at max speed in a micro-centrifuge. The liquid was decanted and the pellet was air dried briefly and resuspended with 8 M Urea, followed by reduction and alkylation with 5 mM DDT and 15 mM idodoacetamide for 30 min at room temperature. Samples were diluted with 100 mM ammonia bicarbonate to final urea concentration of less than 2 M. Afterwards, the samples were digested with 1 µg of trypsin (Promega) at 37 °C overnight. Trypsin was inactivated by 0.1% TFA and samples were desalted by capturing the peptides onto C18 100 μL bed tips (Pierce^®^C18 tips, Thermo Scientific) following the manufacture’s protocol. The bound peptides were eluted with 60% acetonitrile, 0.1% TFA, then dried using a SpeedVac, and resuspended in 20 µL buffer containing 2% acetonitrile with 0.1% acetic acid.

The peptide mixtures from each fraction were sequentially analyzed by liquid chromatography tandem mass spectrometry (LC-MS/MS) using Thermo Ultimate 3000 RSLCnano-Q Exactive mass spectrometry platform nano-LC system (Easy nLC1000) connected to Q Exactive HF mass spectrometer (Thermo Scientific). This platform is configured with nano-electrospray ion source (Easy-Spray, Thermo Scientific), Acclaim PepMap 100 C18 nanoViper trap column (3 μm particle size, 75 μm ID × 20 mm length), EASY-Spray C18 analytical column (2 μm particle size, 75 μm ID × 500 mm length). The data from each sample was collected in triplicate at 2 µL per injection, following which the peptides were eluted at a flow rate of 300 nL/min using linear gradients of 7–25% Acetonitrile (in aqueous phase and 0.1% Formic Acid) for 80 min, followed by to 45% for 25 min, and static flow at 90% for 15 min. The mass spectrometry data was collected in data-dependent manner switching between one full scan MS mode (*m*/*z* 380–1600, resolution 70,000, AGC 3e6) and 10 MS/MS mode (resolution 17,500); where MS/MS analysis of the top 10 target ions were performed once and dynamically excluded from the list for 30 s.

The MS raw data sets were searched against UniProt human database that included common contaminants using MaxQuant software (version 1.5.5.1) [52]. We used default parameters for the searches, first search peptide tolerance 20 ppm, main search peptide tolerance 4.5 ppm, maximum two missed cleavage; and the peptide and resulting protein assignments were allowed at 0.01 FDR (thus 99% confidence level). Protein levels were quantified in 25 FSHD1 patients and 17 healthy controls and reported for each protein as the number of unique peptides detected and the intensity measured. Proteins with altered abundance with greater than 2-fold were selected for further inquiry.

Several pre-processing steps were performed on the raw data values before statistical analysis. Each sample had either 2 or 3 replicates which were averaged to yield a single quantification for each subject for each protein. When a value of zero occurs, it can indicate either a true zero or an assay that did not detect that protein. To accurately reflect protein levels, we incorporated zeroes into our analysis in the following way. If one replicate yielded a zero value, that zero was left as is and treated as a true zero. If two replicates yielded a zero, all values for that protein/sample were set to missing as we cannot distinguish true zeroes from artificial ones. We then applied a normalization factor to the average values to account for differences in the amount assayed per sample. We summed the protein counts for all proteins for each sample and used the maximum value to normalize all other samples. This allowed us to ensure that the amount of proteins assayed were proportional for all samples.

All values were log-transformed for analysis. We assessed the relationship between protein levels and disease severity in the FSHD1 patients using a linear regression model where protein level was the dependent variable, severity was the independent variable, and age and gender were covariates. Regression models were performed only for proteins found in 5 or more samples. Model estimates were reported for each protein and included the coefficient and *p*-value for all terms in the model (severity, age and gender) along with an indication of the direction of each effect. This same method was used to assess the relationship between protein level and the number of D4Z4 repeats. We assessed the difference in protein expression between FSHD1 patients and healthy controls using a linear regression model where protein level was the dependent variable, a categorical indicator of disease was the independent variable, and age and gender were covariates. Again, regression models were performed only for proteins found in 5 or more samples. Model estimates were reported for each protein and included the coefficient and p-value for all terms in the model (disease status, age and gender), an indication of the direction of each effect, and age and gender adjusted means for each disease group. As this part of the analysis was discovery in nature, we did not adjust resulting *p*-values for multiple testing. Our intention was to find those proteins showing some evidence of an effect and to move those proteins forward for an additional evidence-based validation experiment. The significance level for all analyses was set at 0.05.

### 2.7. Enzyme-Linked Immunosorbent Assay (ELISA)

Five proteins were chosen for further validation in a separate set of patients via protein-specific ELISA assays. Human specific protein ELISA kits for human insulin-like growth factor-1 (IGF1) (R&D Systems, Minneapolis, MN, USA), profilin 1 (PFN1) (LSBio, Seattle, WA, USA), S100 Calcium-Binding Protein A8 (S100-A8) (Biotechne, Minneapolis, MN, USA), Proteoglycan 4 (PRG4) (AVIVA Systems Biology, San Diego, CA, USA), Human Tropomyosin alpha-4 chain (TPM4) (MyBioSource, San Diego, CA, USA) were performed to determine protein level in FSHD1 and unaffected controls. Plasma (20 µL) from FSHD1 patients (*n* = 19) and healthy volunteer (*n* = 13) controls (age and gender matched) were tested in duplicate following the manufacturer’s recommended protocols. ELISA values were assessed for normality and a log-transformation applied where appropriate. We assessed the relationship between protein level and severity using, as described above, a linear regression model where protein level was the dependent variable, severity was the independent variable, and age and gender were covariates. We assessed the difference in protein expression between FSHD1 and healthy controls using a linear regression model where protein level was the dependent variable, a categorical indicator of disease was the independent variable, and age and gender were covariates. All analyses were performed at the 0.05 significance level.

## 3. Results

### 3.1. Discovery of Novel Candidate miRNA Biomarkers Associated with FSHD

Sixteen FSHD1 patients with pediatric-onset, matched for sex and age, were selected into two groups of a discovery sample set for circulating biomarker studies: one mild FSHD1 group (*n* = 8), and one severe FSHD1 group (*n* = 8), as determined by an FSHD disease severity score. These two groups were each compared to a group of healthy control volunteers (*n* = 8). Demographics are displayed in Table 1. Patients with severe FSHD1 showed a significantly higher FSHD severity score (12.25 ± 2.76; *p* ≤ 0.00001) than patients with mild FSHD1 (4.88 ± 1.46), with any value of nine or higher being classified as severe FSHD.

Ten miRNAs showed a significant change in expression level in mild FSHD1 plasma versus healthy controls, and 12 miRNAs showed a significant change in expression level in severe FSHD samples versus controls (Table 2). Of these, three miRNAs showed a significant increase in both mild and severe FSHD1 in comparison to healthy controls: miR-32, miR-505, and miR-29b. Each of these three miRNAs showed an approximately two-fold higher change in expression in severe FSHD1 patients than in mild FSHD1 patients versus healthy controls. Of the 19 unique miRNAs identified, several have been previously found to play a role in muscle disease pathways. miR-29b, which is associated with TGFβ-signaling and fibrosis, was up-regulated in both mild and severe FSHD1 patients. Both miR-146b and miR-142-3p, which are known to be up-regulated in inflammatory disease states, were up-regulated in mild FSHD1 patients and have previously been shown to be up-regulated in dystrophinopathy (Becker and Duchenne muscular dystrophy) patients and/or animal models [30,36]. miR-486 has previously been defined as a muscle-enriched microRNA or “myomiR” [53], and was found here to be down-regulated in mild FSHD1 patients (*p* < 0.005).

### 3.2. Bioinformatic Analysis of miRNA Regulation and Pathways

To examine their regulation by transcription factors which are dysregulated by the FSHD disease process, we next performed bioinformatic analyses of ChIP-seq data for DNA binding by transcription factors in proximity to each candidate miRNA’s genomic locus. To gain insight into direct consequences of *DUX4*-up-regulating mutations that cause FSHD, we analyzed ChIP-seq data for DUX4. To do this, we analyzed DUX4 binding via a user-supplied DUX4 ChIP-seq track published by Geng et al. [9]. Genes for 16 of the candidate miRNAs had at least one binding site within distances capable of providing gene enhancer functions (Figure 1). Examination of the miR-100 home gene (*MIR100HG*) locus was particularly interesting. In total, we found 18 DUX4 binding sites in the area surrounding *MIR100HG*, and many of these clearly overlapped with histone modifications associated with active promoters (H3K4 tri-methylation) and regulatory elements (H3K27Ac). These data are consistent with regulation of miR-100 expression by DUX4 (Figure 1b). 

To gain insight into additional pathways that may drive expression of candidate miRNAs and contribute to FSHD molecular pathophysiology, we performed bioinformatic analyses of ChIP-seq data for transcription factors that are dysregulated as a result of DUX4 mutations. For this, we obtained a list of transcription factors which are expressed at significantly different levels in human skeletal muscle cells as a result of DUX4 overexpression [9]. We then queried publicly available ChIP-seq datasets to identify which of these transcription factors had ChIP-seq datasets available through the ENCODE public research consortium [46,47]. Of the transcription factors in this dataset, 34 had ChIP-seq datasets available in the Factorbook repository and 47 had ChIP-seq datasets available in the ENCODE 3 repository [48,49,50]. Genomic binding by each of these transcription factors was surveyed for each of these transcription factors for all candidate miRNAs (Appendix A). Transcription factors that increased in response to overexpression of toxic, full-length DUX4 but did not increase in response to a non-toxic, truncated isoform of DUX4 were considered to be of particular interest (Figure 2a). Of these factors, four showed a particularly high number of binding sites within regulatory distance of the candidate miRNAs: early growth response protein 1 (EGR1), FOS, MYC, and yin yang 1 (YY1). As an example of these findings, miR-576 was up-regulated in FSHD patients, has five DUX4 binding sites neighboring its home gene (SEC24B), and has a high number of binding sites for the secondary transcription factors described here (Figure 2b). EGR1, FOS, MYC and YY1 all showed a large number of binding sites around miR-576, and these frequently overlapped with histone modifications which mark active promoter and enhancer regions, consistent with these four transcription factors driving gene expression signatures in FSHD.

Additionally, we used Ingenuity Pathway Analysis software to perform a bioinformatic analysis on the candidate miRNAs identified in this study, together with transcription factors previously published to be dysregulated in FSHD [9], to see if there are defined signaling pathways or interactions shared by these factors. Interestingly, this analysis showed that there are previously established connections between many of the miRNAs and transcription factors examined, with 15 of the miRNAs and 18 of the transcription factors found to make up a network with previously defined interactions (Figure 3). For example, increased levels of miR-34a are known to decease cAMP response element-binding protein (CREB) to drive neuronal dysfunction in HIV-induced neurocognitive disorders, and to increase AMP-dependent transcription factor 3 (ATF3) levels in colon cancer [77,78]. MYC binds to ATF3 as well as to lysine-specific demethylase 5B (KDM5B) and YY1, all four of which are elevated in FSHD [9,79,80,81]; in addition, MYC is known to activate transcription of both enhancer of zeste homolog 2 (EZH2) and miR-9 [82,83], both of which are also increased in FSHD. Together, these bioinformatics data show our candidate miRNA markers are consistent with a change in transcriptional programming that results from FSHD-causing DUX4 overexpression mutations.

### 3.3. Confirmation of miRNA Increases in FSHD1 Patients

Next, we assayed expression of candidate miRNA biomarkers in samples from a separate and non-overlapping group of patients. Upon clinical examination, all patients in this validation group were determined to have FSHD1. We selected 14 miRNAs that significantly increased in the discovery experiments for follow-up study in the validation group. We found three of these miRNAs (miR-9, miR-32 and miR-329) were not expressed at consistently high enough levels for detection within plasma from the validation set of FSHD1 patients, leaving 11 miRNAs for validation. Here, these 11 individual candidate miRNAs were quantified in FSHD1 (*n* = 12; 9 females, 3 males) versus healthy volunteer control samples (*n* = 7; 4 females, 3 males).

Upon quantification, we found 8 of these 11 candidate miRNAs also showed a clear increase in samples from the FSHD validation group in comparison to healthy controls (Figure 4). miR-100, miR-103, miR-29b, miR-34a, miR-454, miR-505 and miR-576 were all expressed at significantly higher levels (*p* ≤ 0.05) in FSHD1 serum. miR-100, miR-29b, miR-34a, miR-505, and miR-576 were the most highly up-regulated in FSHD1, showing up-regulation from approximately 4- to 20-fold higher than healthy controls. miR-146b was also expressed at an approximately 2-fold higher level in this set of FSHD1 patients; however it did not reach significance (*p* = 0.06). Of the remaining three miRNA candidates, miR-98 showed no apparent change, while miR-141 and miR-142-3p showed an approximately 50% increase that did not reach significance. As most candidate miRNAs showed consistent behavior in this separate validation set of FSHD samples, this panel of miRNAs merits further investigation as biomarkers moving forward.

### 3.4. Proteomics Profiling

To identify protein candidate biomarkers, we performed LC-MS/MS-based proteomic profiling of samples from a discovery group of FSHD patients (Table 3). For this, plasma from FSHD1 patients (*n* = 25) was compared to healthy volunteer controls (*n* = 17), with a roughly even mix of males and females, and an average age of early- to mid-twenties for each group. All FSHD patients were confirmed to have FSHD1 resulting from D4Z4 contraction mutations that alter epigenetic regulation of DUX4.

Based on signal intensity, we identified 32 proteins that were significantly different between FSHD1 and healthy control samples (Appendix A). To further filter the protein list, we used unique peptide count data to identify proteins that had significantly different counts between FSHD1 and control samples. This narrowed the candidates down to 13 proteins (Table 4). Within these protein markers, fibulin-1 (FBLN1) and insulin-like growth factor 1 (IGF1) showed potential effects of sex and age, while keratin 16 (KRT16) displayed a potential age effect and profilin-1 (PFN1) showed a potential sex effect (Appendix A). Among the 13 total protein biomarker candidates, 11 proteins were higher in FSHD1 samples versus healthy controls, while two proteins were lower in the FSHD1 samples versus healthy controls.

We selected five candidate protein markers for subsequent quantification via protein-specific ELISA analysis of a non-overlapping validation group of FSHD1 samples. These included insulin-like growth factor 1 (IGF1), proteoglycan 4 (PRG4), profilin 1 (PFN1), tropomyosin 4 (TPM4), and S100 calcium-binding protein A8 (S100A8). Of these candidate proteins, S100A8 showed a significant increase in FSHD1 plasma of approximately 4.5-fold over healthy controls in the validation group (Figure 5a), consistent with its behavior in the discovery experiment. To determine if elevated S100A8 signaling was consistent with the overall proteomic and miRNA profiling results, we performed bioinformatic pathway analyses focused on the S100A8 pathway along with the full list of candidate protein (Figure 5b) and miRNA (Figure 5c) markers. Nine proteins and 13 miRNAs were shown to have previously established connections to the toll-like receptor 4 (TLR4) signaling pathway, which is activated by S100A8 and drives increased inflammatory (NF-κB and AP-1) gene expression. As miRNAs can reflect a direct readout of transcription factor activity, we also surveyed ChIP-seq data to analyze DNA regions encoding miRNAs elevated in FSHD for binding by the NF-κB and AP-1 transcription factors activated by S100A8 (Figure 5d). All miRNAs except for one (miR-329) showed binding by NF-κB and/or AP-1 subunits at DNA regions capable of acting as regulatory promoter or enhancer elements. As S100A8 is a well-established biomarker of inflammatory disease processes (reviewed in [86]) and these can be up-regulated in the muscular dystrophies, this protein merits further investigation as a biomarker for FSHD moving forward.

## 4. Discussion

There is currently no effective treatment available for FSHD. However, research advances in FSHD are now beginning to yield promising and novel therapeutic strategies that will require well-designed clinical trials to evaluate effectiveness. Potential therapeutic strategies including antisense oligonucleotides (AON) and small molecules have been reported or are being actively pursued [12,97,98,99,100]. Changes in biomarkers following a treatment can be a powerful tool for evaluating the efficacy and safety of the treatment. Previous studies seeking to identify circulating miRNA biomarkers in muscular dystrophy have focused exclusively on assaying myomiRs, which are a defined group of miRNAs with muscle-specific or muscle-enhanced expression [101,102]. Previously, a study by Statland et al. identified 7 potential protein biomarkers in 22 FSHD serum samples, using a commercial multiplex assay [103]. A multi-site study using aptamer-based SomaScan proteomics to assay two FSHD populations identified a total of 115 proteins that were dysregulated, four of which behaved consistently between the two independent cohorts (creatine kinase MM, creatine kinase MB, carbonic anhydrase III, and troponin I type 2) [104]. In this study, we used -omics approaches to identify additional circulating miRNA and protein biomarker candidates using samples collected from individuals with early-onset FSHD.

There is an intriguing potential for developing miRNAs as biomarkers in diseases affecting muscle, as they are stable in biofluids, objective, minimally invasive, and well-conserved between human patients and preclinical animal models [23,24]. Recently, the utility of serum miRNAs to detect muscle involvement in complex diseases with highly variable symptoms has been demonstrated, as in patients with MADD [25]. Muscle-specific miRNAs are also elevated in Duchenne and Becker muscular dystrophy, along with a set of inflammatory miRNAs reflecting the chronic inflammatory pathology of these diseases [29,30,105]. Here we identify eight circulating miRNAs that are associated with FSHD in patient plasma samples. The prevalence of DNA binding by DUX4 and FSHD-associated transcription factors, within regions capable of regulating the candidate miRNAs, provides a molecular rationale for their up-regulation in FSHD. Several of the markers have also been previously shown to play a role in muscle diseases and associated pathological pathways. These candidate biomarkers hold potential as monitoring biomarkers in early-onset FSHD.

Several candidate miRNAs we identified have previously been proposed as circulating biomarkers and have shown similar behavior in other diseases. Plasma miR-454 has been identified as a biomarker of myotonic dystrophy [72,73]. Serum miR-146b is a pharmacodynamic biomarker in inflammatory bowel disease (IBD) [34,35]. Intriguingly, miR-146b is also known to down-regulate dystrophin in multiple muscle diseases, is increased in dystrophinopathies and in myositis, and is also drug-responsive in the *mdx* mouse model of DMD [30,31]. Urinary miR-141 provides a promising diagnostic biomarker for the identification of both prostate and bladder cancers [69]; it will be interesting to determine if this or other candidate miRNAs are also dysregulated in urine from dystrophic patients, as this sampling method could provide a completely non-invasive biomarker.

Increases in circulating S100A8, a subunit of calprotectin, are consistent with an inflammatory signature playing a role in FSHD. The inflammatory calprotectin protein consists of a heterodimer (S100A8/S100A9) which binds to toll-like receptor 4 (TLR4) to activate pro-inflammatory gene expression pathways through the NF-κB and AP-1 transcription factors. Consistent with such an inflammatory gene signature in FSHD, bioinformatic analyses here show five of the candidate miRNAs have established connections with TLR4 signaling, are increased in FSHD patients, and have gene promoters that are bound by AP-1 and/or NF-κB. Outside of FSHD, calprotectin is already a well-established biomarker across rheumatic diseases. Fecal calprotectin is a widely used diagnostic, monitoring and pharmacodynamic biomarker for IBD, and recent studies indicate serum calprotectin levels are also well-correlated with IBD disease state [87,106]. Serum calprotectin is used as a monitoring and pharmacodynamic biomarker for rheumatoid arthritis, and intriguingly S100A8/S100A9 may have further utility in arthritis as a molecular imaging marker of inflammatory activity [84,107,108]. Of particular relevance to the present study, calprotectin in both muscle and serum is a biomarker for disease activity in juvenile dermatomyositis [109]. Moving forward, it will be interesting to see if S100A8 or calprotectin can show further utility as completely non-invasive or local biomarker for FSHD and other muscle diseases such as myositis.

Several of the molecular markers we identified here as elevated in FSHD may provide a new therapeutic target. In various states of muscle atrophy miR-29b is also up-regulated, while preventing its expression shows efficacy in mouse models of muscle atrophy [64,65]. In myositis and Becker muscular dystrophy, the inflammatory marker miR-146b is known to down-regulate dystrophin expression, whereas the reduction of miR-146b via anti-inflammatory drugs or via miRNA-targeting oligos is proposed as a method to increase dystrophin levels to help improve muscle health [30,31]. In various rheumatological disease states, the inhibition of S100A8 or calprotectin via small molecule inhibitors or antibodies is a very attractive therapeutic strategy; early studies of such inhibitors are already showing therapeutic efficacy in both human trials and/or in mouse models, including in studies for arthritis, asthma, IBD, and multiple sclerosis (reviewed in [86]). Similarly, decreases in PROC seen here in FSHD are also seen in several rheumatological disorders, where treatment with PROC activators are already being pursued as a therapeutic option (reviewed in [94]).

Bioinformatic analyses of the -omics results support muscle and inflammatory gene expression pathways as being dysregulated in FSHD. As expected, several muscle pathology-associated miRNAs are dysregulated in FSHD patients: miR-486 is a defined myomiR, miR-29b up-regulation promotes muscle atrophy, miR-146b is dysregulated in dystrophinopathies and myositis, miR-329 counteracts muscle hypertrophy, and three others are known to be dysregulated in myotonic dystrophy, lamin A (LMNA) dystrophy, and/or FSHD (miR-34a, miR-140-3p, miR-100, and miR-454). Consistent with these findings, several of the proteins that were dysregulated are known to function in muscle contraction, actin filament organization and/or muscle regeneration (TOM4, PFN1, CFL1, TMSB4X, and IGF1).

S100A8 and its associated inflammatory signaling pathway (TLR4, NF-κB and AP-1) appear to be a substantial hub for dysregulated expression of the candidate markers we identified. Nine of the candidate miRNAs have previously established connections to this TLR4-centered pathway. ChIP-seq analysis of the miRNAs up-regulated in FSHD shows all but one have promoters bound by NF-κB or AP-1, which are activated by S100A8-induced TLR4. In the proteomics data, several of the proteins that increased are pro-inflammatory (S100A8, KRT16 and SPP2) while in contrast the two proteins that decreased have anti-inflammatory (PROC and PRG4) roles. Consistent with our FSHD findings, KRT16 and S100A8 are also up-regulated together in inflammatory skin disorders; additionally, the pattern of increased S100A8 with decreased PROC is seen here in FSHD as well as in IBD and several other chronic inflammatory disorders [93,94]. Pathway analysis further establishes a link between the protein markers, as nine out of 14 have established connections to the S100A8 and TLR4 signaling pathway. Together these data confirm that circulating FSHD biomarkers reflect muscle pathogenesis, and suggest inflammatory S100A8/TLR4 signaling plays a role in pediatric-onset FSHD as well.

## 5. Conclusions

FSHD is chronic genetic muscle disease with a variable prognosis. There is no cure, and no pharmaceuticals for FSHD have shown efficacy in altering the disease course. Development of objective biomarkers will facilitate the clinical and preclinical development of novel therapies, as well as our ability to monitor disease activity. We identified eight circulating miRNAs (miR-100, miR-103, miR-146b, miR-29b, miR-34a, miR-454, miR-505, and miR-576) which may be developed as biomarkers for FSHD. Additionally, we identified the S100A8 subunit of calprotectin as a primary protein marker of interest for FSHD, consistent with its utility in numerous rheumatic diseases. These molecular markers warrant further investigation in additional cohorts, preclinical drug testing, and prospective clinical trials.

## Figures and Tables

**Figure 1 jpm-10-00236-f001:**
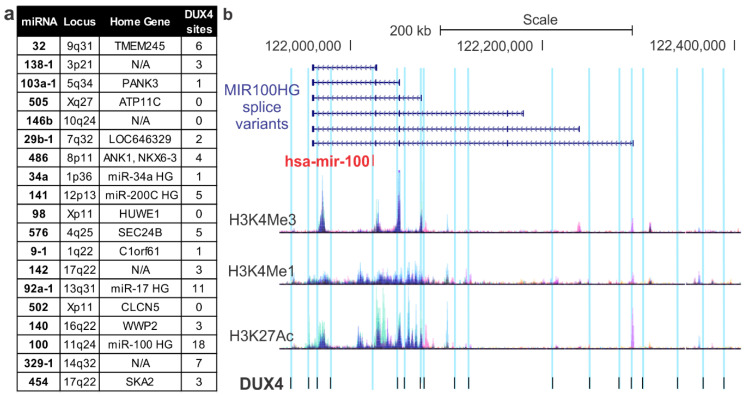
DUX4 binding sites at loci surrounding miRNAs dysregulated in FSHD patients. The 19 miRNAs dysregulated in FSHD1 patient plasma samples were queried for potential regulation by the DUX4 transcription factor, which aberrantly expressed in FSHD, using a DUX4 ChIP-seq dataset [9]. (**a**) Overview of all DUX4 binding sites within regions capable of acting as regulatory elements (100 kb) of the 19 miRNAs and their home genes. (**b**) Schematic of DUX4 binding sites within the miR-100 locus and its surrounding home gene (*MIR100HG*) variants. Note, miR-100 is transcribed from right to left on this image. Corresponding epigenetic modification maps display the location of histone modifications associated with active promoters (H3K4me3) and poised/active enhancers (H3K4me1 and H3K27Ac, respectively).

**Figure 2 jpm-10-00236-f002:**
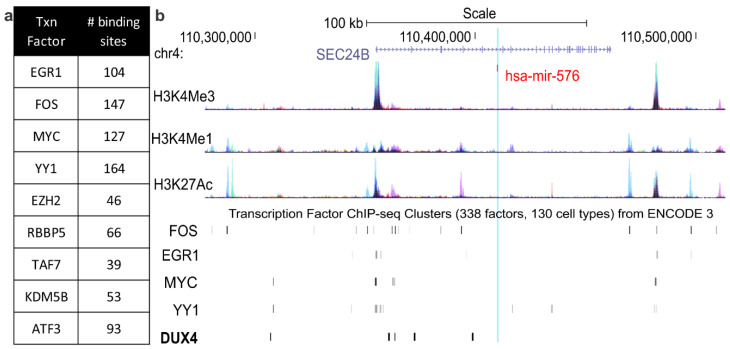
Candidate miRNA loci are consistent with regulation via transcription factors dysregulated in FSHD. (**a**) Table listing a subset of transcription factors which are each increased in human skeletal muscle cells in response to DUX4 overexpression [9], along with the number (#) of binding sites they show within potential regulatory distance (100 kb) of the 19 candidate miRNAs. (**b**) The miR-576 locus shows binding consistent with regulation by FOS, EGR1, MYC, YY1, and DUX4. Corresponding epigenetic modification maps display the location of histone modifications associated with active promoters (H3K4me3) and poised/active enhancers (H3K4me1 and H3K27Ac) in the vicinity of the miR-576 locus and its surrounding home gene, *SEC24 homolog B* (*SEC24B*). (DUX4 binding sites identified using ChIP-seq data uploaded from Geng et al. [9]; binding sites for additional transcription factors identified using UCSC Genome Browser and respective ChIP-seq datasets accessed via the ENCODE3 regulation track [46,47,48,49,50]).

**Figure 3 jpm-10-00236-f003:**
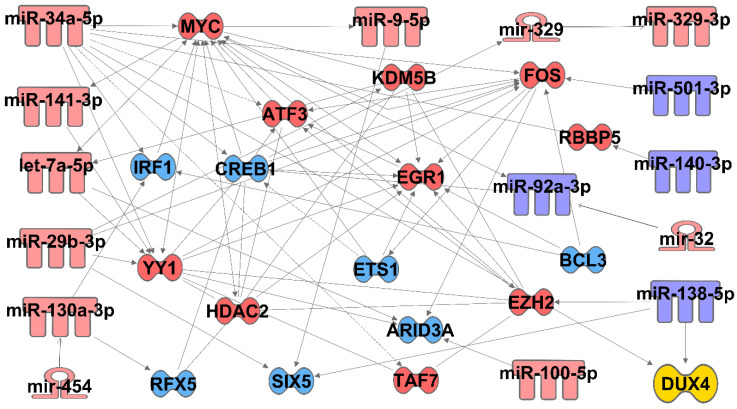
Pathway analysis of miRNAs and transcription factors dysregulated by FSHD mutations. Ingenuity Pathway Analysis software was used to identify established connections between candidate miRNAs from this study with transcription factors known to be dysregulated by FSHD-causing overexpression of DUX4 [9]. Red-shaded miRNAs and transcription factors were observed to increase, while those shaded blue were observed to decrease. Solid arrows denote direct relationships, while dashed arrows denote indirect relationships.

**Figure 4 jpm-10-00236-f004:**
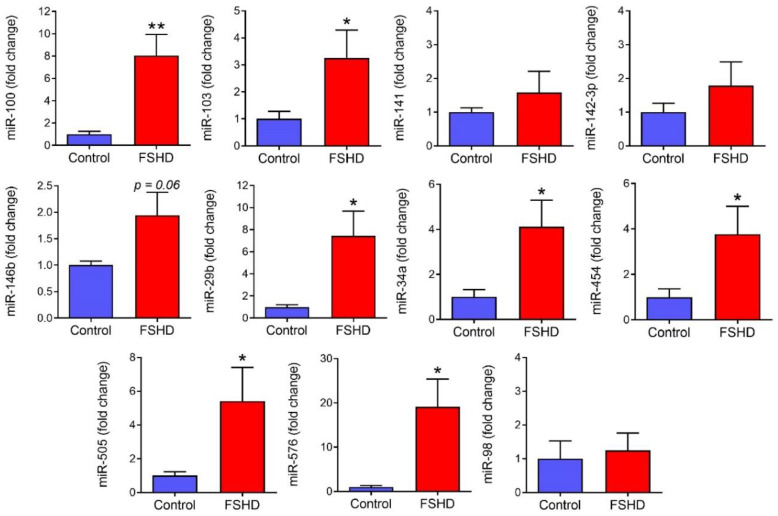
Expression of candidate miRNAs in a validation group of FSHD patients. Candidate miRNAs that increased in the FSHD discovery experiment were assayed via individual qRT-PCR assay in a separate validation group of FSHD1 patient plasma samples. Expression levels of each miRNA are expressed as fold change versus healthy control volunteers. (values are mean ± SEM, * *p* ≤ 0.05, ** *p* ≤ 0.01, one-tailed *t*-test comparing FSHD1 to control in direction of Discovery experiment; one outlier removed from miR-34a and miR-576 after significant Grubb’s outlier test; *n* = 7 healthy control volunteers, 12 FSHD1).

**Figure 5 jpm-10-00236-f005:**
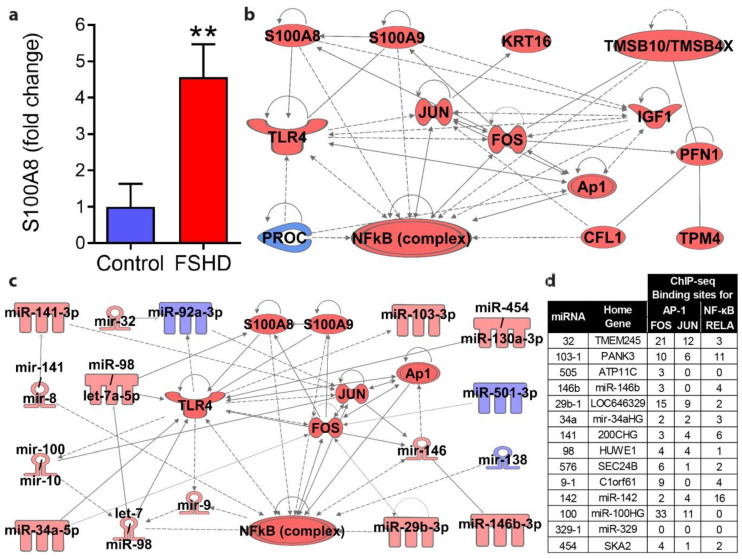
Validation and pathway analysis of elevated S100A8 protein in FSHD. (**a**) ELISA of S100A8 protein in plasma from a separate validation set of FSHD1 patients. (**b**) Bioinformatic pathway analysis was used to identify known connections between candidate protein markers with S100A8 pathway proteins involved in TLR4 signaling. (**c**) Bioinformatic pathway analysis was used to identify established connections between candidate miRNAs with S100A8 pathway proteins involved in TLR4 signaling. (**d**) Bioinformatic analysis of ChIP-seq defined binding sites for the key S100A8 pathway transcription factors AP-1 (FOS and JUN) and NF-κB (RELA), at potential regulatory regions of the candidate miRNAs that were found to increase in FSHD plasma. Binding sites represent the combined number of potential promoter (within 2 kb of promoter) and enhancer (within 10 kb) regulatory regions with ChIP-seq-confirmed transcription factor binding for each miRNA home gene. (** *p* ≤ 0.01; *n* = 13 healthy control volunteers, 19 FSHD1; panels (**b**,**c**) produced using Ingenuity Pathway Analysis software, red = increased, blue = decreased; data for panel (**d**) produced using the Factorbook ChIP-seq data repository from ENCODE and the UCSC genome browser).

**Table 1 jpm-10-00236-t001:** Clinical characteristics of study group patients.

	Healthy Control	Mild FSHD	Severe FSHD
**N**	8	8	8
**Age in years (mean ± SD)**	28.29 ± 15.82	24.84 ± 10.46	27.58 ± 15.11
**Males:Females**	4:4	4:4	4:4
**FSHD severity score**	N/A	4.88 ± 1.46	12.25 ± 2.76 **

** *p* ≤ 0.00001, *t*-test of mild FSHD versus severe FSHD severity score.

**Table 2 jpm-10-00236-t002:** Discovery of 19 circulating miRNAs with altered expression in mild or severe FSHD.

Mild FSHD Versus Healthy Controls				
miRNA	↑ or ↓	*p*-Value	Rq *	Known Roles in Muscle/Disease Pathways
138	↓	0.004	0.05	Heart development; hypoxia and S100A1 [54,55,56]
486	↓	0.009	0.26	myomiR; steroid-response in IBD blood [35,53]
9	↑	0.017	9.58	Inhibits satellite cells; COPD weakness [57,58]
32	↑	0.020	8.45	Cardiac fibrosis; VSMC calcification [59,60]
146b	↑	0.034	2.18	Up-regulated in DMD and BMD [30,36]
92a	↓	0.039	0.31	Inhibits myogenic differentiation via Sp1 [61]
576	↑	0.043	3.64	Up-regulated in smooth muscle tumors [62]
142-3p	↑	0.044	2.69	Elevated in models of DMD and myositis [31,36]
505	↑	0.046	9.69	Cardiac development and regeneration [63]
29b	↑	0.050	17.48	Muscle atrophy, therapeutic target [64,65]
**Severe FSHD versus Healthy Controls**				
*32*	*↑*	*0.001*	*17.09*	*Cardiac fibrosis; VSMC calcification* [59,60]
*505*	*↑*	*0.007*	*19.51*	*Cardiac development and regeneration* [63]
502-3p	↓	0.009	0.36	Myogenic differentiation; ACAD marker [66,67]
103	↑	0.013	4.29	Myogenic differentiation [67]
98	↑	0.014	21.65	Muscle differentiation [68]
141	↑	0.016	7.52	Biomarker for prostate and bladder cancer [69]
*29b*	*↑*	*0.018*	*28.78*	*Muscle atrophy, therapeutic target* [64,65]
34a	↑	0.024	8.12	Up in FSHD and myotonic dystrophy [70,71]
140-3p	↓	0.028	0.54	Plasma biomarker of myotonic dystrophy [72,73]
100	↑	0.029	3.58	Up-regulated in LMNA dystrophy biopsies [74]
329	↑	0.030	4.63	Counteracts muscle hypertrophy [75]
454	↑	0.046	2.02	Plasma biomarker of myotonic dystrophy [72,73]
**Severe FSHD versus Mild FSHD**				
502-3p	↓	0.041	0.45	Myogenic differentiation; ACAD marker [66,67]
95	↑	0.042	2.21	Up in DMD patient and dog model serum [76]
886-3p	↑	0.048	3.27	Up in plasma of myotonic dystrophy patients [73]

*Italics* = dysregulated in both mild and severe FSHD; ACAD = acute coronary artery disease, BMD = Becker muscular dystrophy, COPD = chronic obstructive pulmonary disease, DMD = Duchenne muscular dystrophy, IBD = inflammatory bowel disease, LMNA = Lamin A/C, TGFβ = Transforming Growth Factor β, VSMC = vascular smooth muscle cell. * *p* < 0.005.

**Table 3 jpm-10-00236-t003:** Clinical characteristics of patients in proteomics discovery group.

	Healthy Control	FSHD
**N**	17	25
**Age in years (mean** **± SD)**	23.45 ± 13.18	25.68 ± 14.71
**Males:Females**	9:8	13:12
**FSHD severity score**	N/A	8.54 ± 4.10

**Table 4 jpm-10-00236-t004:** Thirteen circulating proteins identified as dysregulated in FSHD plasma via LC-MS/MS.

Gene Name	UniProt ID	↑ or ↓	*p*-Value	Known Roles in Muscle/Disease
F13A1	P00488	↑	0.031	Hypertension, angiotensin II, coagulation
IGF1	P05019	↑	0.043	hypertrophy, development, satellite cells, regeneration
S100A8	P05109	↑	0.009	TLR4; pro-inflammation, up in rheumatic diseases [84,85,86,87]
PFN1	P07737	↑	0.010	actin cytoskeleton organization
FBLN1	p23142	↑	0.011	positive regulation of fibroblast proliferation
CFL1	P23528	↑	0.031	actin filament organization and depolymerization
TMSB4X	P62328	↑	0.017	actin filament organization
TPM4	P67936	↑	0.015	actin organization, muscle contraction
EFEMP1	Q12805	↑	0.001	plasma biomarker for mesothelioma; retinal dystrophy [88]
KRT16	P08779	↑	0.009	elevated with S100A8 in skin disorders, psoriasis [85,89,90,91]
SPP2	Q13103	↑	0.017	pro-inflammatory, NF-κB; blood pressure; bone health [92]
PROC	P04070	↓	0.048	anti-inflammatory, down in chronic inflammation [93,94]
PRG4	Q92954	↓	0.024	TLR4; anti-inflammatory, down in arthritis [95,96]

CFL1 = Cofilin 1, EFEMP1 = EGF-containing fibulin-like extracellular matrix protein 1, F13A1 = Coagulation factor XIII A chain, FBLN1 = fibulin-1, IBD = inflammatory bowel disease, IGFI = insulin-like growth factor 1, KRT16 = Keratin 16, PFN1 = Profilin-1, PRG4 = Proteoglycan 4 or lubricin, PROC = Protein C, S100A8 = S100 calcium-binding protein A8, SPP2 = Secreted phosphoprotein 24, TLR4 = Toll-like receptor 4, TMSB4X = Thymosin beta-4, TPM4 = Tropomyosin alpha-4 chain.

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
