# Peer review of "Multi-Omics Identifies Circulating miRNA and Protein Biomarkers for Facioscapulohumeral Dystrophy"

_jpm, 2020, doi:10.3390/jpm10040236_

Round 1

Reviewer 1 Report

The paper from Christopher R. Heier, et al. was well designed and implemented, but some more recent references are needed in both introduction and discussion. A few examples follows:

Missaglia S, Pegoraro V, Marozzo R, Tavian D, Angelini C. Correlation between ETFDH mutations and dysregulation of serum myomiRs in MADD patients. Eur J Transl Myol. 2020 Apr 1;30(1):8880. doi: 10.4081/ejtm.2019.8880. eCollection 2020 Apr 7.

Roberta Marozzo , Valentina Pegoraro and Corrado Angelini. MiRNAs, Myostatin, and Muscle MRI Imaging as Biomarkers of Clinical Features in Becker Muscular Dystrophy. Diagnostics 2020, 10(9), 713; https://doi.org/10.3390/diagnostics10090713

Ngoc-Lan Nguyen , Can Thi Bich Ngoc , Chi Dung Vu , Thi Thu Huong Nguyen and Huy Hoang Nguyen. Whole Exome Sequencing as a Diagnostic Tool for Unidentified Muscular Dystrophy in a Vietnamese Family.Diagnostics 2020, 10(10), 741; https://doi.org/10.3390/diagnostics10100741    

Author Response

The paper from Christopher R. Heier, et al. was well designed and implemented, but some more recent references are needed in both introduction and discussion. A few examples follows:

Missaglia S, Pegoraro V, Marozzo R, Tavian D, Angelini C. Correlation between ETFDH mutations and dysregulation of serum myomiRs in MADD patients. Eur J Transl Myol. 2020 Apr 1;30(1):8880. doi: 10.4081/ejtm.2019.8880. eCollection 2020 Apr 7.

  • We have now added this and other new references to the Introduction and to the Discussion.Text added to Introduction line 92:
    • “Recently, circulating miRNAs have emerged as exciting potential diagnostic, prognostic, and drug-responsive biomarkers. This is a class of small non-coding RNA molecules (~22 nucleotides in length) that can help to regulate gene expression [22], and which are highly stable in biofluids such as blood and urine [23,24]. In rare diseases with highly variable symptoms, such as multiple acyl-coenzyme A dehydrogenase deficiency (MADD), the serum-based detection of muscle-specific miRNAs termed myomiRs can signal the presence of underlying muscle-specific pathologies [25]. In Duchenne and Becker muscular dystrophies, myomiRs are upregulated in serum from both patient populations, while detection of miR-206 upregulation can be used to differentially diagnose severe Duchenne versus Becker patients [26-28]. In addition to myomiRs, inflammatory miRNAs such as miR-146a, miR-146b, miR-221 and miR-155 have been found to be dysregulated in multiple forms of muscular dystrophies [29-31]. These two classes of miRNA show potential as pharmacodynamic biomarkers, with myomiRs proposed for muscle-stabilizing treatments such as gene therapy [32,33], and inflammatory microRNAs proposed for current steroids [34,35] as well as newly emerging dissociative anti-inflammatory drugs such as vamorolone [36-38] or edasalonexent [39,40]. In parallel to development of miRNA monitoring biomarkers, new advances in whole exome sequencing are enabling clinicians to diagnose novel mutations in over 60 genes known to be responsible for muscular dystrophies such as FSHD and limb-girdle muscular dystrophy (LGMD) [41-44]. Together, these advances will help to improve the diagnosis, monitoring, and treatment of a diverse number of diseases affecting muscle.”
    • Text added to Discussion:
      • “Recently, the utility of serum miRNAs to detect muscle involvement in complex diseases with highly variable symptoms has been demonstrated, as in patients with multiple acyl-coenzyme A dehydrogenase deficiency (MADD) [25]. Muscle-specific miRNAs are also elevated in Duchenne and Becker muscular dystrophy, along with a set of inflammatory miRNAs reflecting the chronic inflammatory pathology of these diseases [29,30,104].”

Roberta Marozzo , Valentina Pegoraro and Corrado Angelini. MiRNAs, Myostatin, and Muscle MRI Imaging as Biomarkers of Clinical Features in Becker Muscular Dystrophy. Diagnostics 2020, 10(9), 713; https://doi.org/10.3390/diagnostics10090713

  • We have now added this and other new references to the Introduction and to the Discussion. See above for text.

Ngoc-Lan Nguyen , Can Thi Bich Ngoc , Chi Dung Vu , Thi Thu Huong Nguyen and Huy Hoang Nguyen. Whole Exome Sequencing as a Diagnostic Tool for Unidentified Muscular Dystrophy in a Vietnamese Family.Diagnostics 2020, 10(10), 741; https://doi.org/10.3390/diagnostics10100741    

  • We have now added this and other new references to the Introduction and to the Discussion. See above for text.

Reviewer 2 Report

In this manuscript, Heier et al. perform miRNA and proteomic profiling of plasma samples from Fascioscapulohumeral Dystrophy Type 1 (FSHD1) patients (mild and severe) to identify potential disease biomarkers. They identify 19 miRNAs and 13 proteins whose expression is significantly altered in either group of patients compared to controls. The authors then validate the bioinformatically identified candidate miRNAs and proteins in a distinct group of FSHD1 samples and find that the expression of 8 of the candidate miRNAs and 1 of the candidate proteins is significantly altered in FSHD1 patients compared to controls. In addition, the authors used ChIP-seq databases to determine the potential association of the miRNAs with transcription factors pathologically affected in FSHD1, supporting an overall change in transcriptional programming. Further bioinformatic analyses of the candidate miRNAs and proteins reveal an association with the toll-like receptor-4 pathway. Overall, this is a well written manuscript that offers novel insights on potential biomarkers and pathological signalling pathways in FSHD that could could be valuable for pre-clinical and clinical endeavours.

Comments/Suggestions:

  1. Table 2: Were there any miRNAs that were different between mild and severe FSHD1? If so, please provide those in the table as well.
  2. Results Section 3.2: Whilst it is clear in the methods section that the ChIP-seq data was not acquired herein but obtained from an available database (ENCODE), this should be re-specified in results section.
  3. Lines 340-342: Here again, please also ensure that it is clear that the miRNAs are those obtained in the experimental study and the transcription factors obtained from the database.
  4. Line 342: "this this" typo.
  5. Line 345: Perhaps expand on a few of these previously identified interactions directly in the text.
  6. In both the miRNA and protein profiling data, did the authors observe any distinct patterns when performing sex-specific or more age-constrained comparisons?

Author Response

1. Table 2: Were there any miRNAs that were different between mild and severe FSHD1? If so, please provide those in the table as well.
- We have now compared mild and severe FSHD1 data, and identified 3 miRNAs that were significantly different between these two groups. These have been added to Table 2 as recommended.
2. Results Section 3.2: Whilst it is clear in the methods section that the ChIP-seq data was not acquired herein but obtained from an available database (ENCODE), this should be re-specified in results section.
- We have now re-specified this in the Results section:
1. Line 312: “ To do this, we analyzed DUX4 binding via a user-supplied DUX4 ChIP-seq track published by Geng et al [9].”
2. Line 334: “We then queried publicly available ChIP-seq datasets to identify which of these transcription factors had ChIP-seq datasets available through the ENCODE public research consortium [46,47]. Of the transcription factors in this dataset, 34 had ChIP-seq datasets available in the Factorbook repository and 47 had ChIP-seq datasets available in the Encode 3 repository [48-50].”
3. Lines 340-342: Here again, please also ensure that it is clear that the miRNAs are those obtained in the experimental study and the transcription factors obtained from the database.
- We have clarified this in the Results:
1. “we used Ingenuity Pathway Analysis software to perform a bioinformatic analysis on the candidate miRNAs identified in this study, together with transcription factors previously published to be dysregulated in FSHD [9], to see if there are defined signaling pathways or interactions shared by these factors.”
4. Line 342: "this this" typo.
- We have corrected this typo.
5. Line 345: Perhaps expand on a few of these previously identified interactions directly in the text.
- We have now expanded on a few of the previously identified interactions in the text in section 3.2:
1. “For example, increased levels of miR-34a are known to decease CREB to drive neuronal dysfunction in HIV-induced neurocognitive disorders, and to increase ATF3 levels in colon cancer [77,78]. MYC binds to ATF3 as well as to KDM5B and YY1, all four of which are elevated by FSHD-causing mutations [9,79-81]; in addition, MYC is known to activate transcription of both EZH2 and miR-9 [82,83], both of which are also increased in FSHD.”
6. In both the miRNA and protein profiling data, did the authors observe any distinct - patterns when performing sex-specific or more age-constrained comparisons?
- We observed potential sex and age effects for FBLN1, IGF1, KRT16 and PFN1. We have now added this to the results and as a Supplemental Table S3. Text added to line 402:
1. “Within these protein markers, fibulin-1 (FBLN1) and insulin-like growth factor 1 (IGF1) showed potential effects of sex and age, while keratin 16 (KRT16) displayed a potential age effect and profilin-1 (PFN1) showed a potential sex effect (Table S3).”